# Effect of Housing Support Programs on Residential Satisfaction and the Housing Cost Burden: Analysis of the Effect of Housing Support Programs in Korea Based on Household Attributes

Saehim Kim 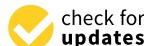, Joonwon Hwang and Myeong-Hun Lee *

Department of Urban and Regional Development, Graduate School of Urban Studies, Hanyang University, 222 Wangsimni-ro, Seongdong-gu, Seoul 04763, Korea
* Correspondence: mhlee99@hanyang.ac.kr; Tel.: +82-2-2220-4418

**Abstract:** Korea is implementing housing support programs such as public rental housing and housing allowances to improve the housing welfare of low-income households. In this study, we empirically analyzed the effects of the public rental housing program and the housing allowance program on residential satisfaction and the housing cost burden of policy beneficiaries. In accordance with household attributes, we analyzed how the status of using these programs affected each group's residential satisfaction and housing cost burden. We used the data from the 2020 Korea Housing Survey conducted by the Ministry of Land, Infrastructure and Transport. We examined the housing support programs' effects on each of the following household groups: all households, one-person households, households of newlywed couples, young adult households, and households of the elderly. The status of residing in public rental housing positively affected the residential satisfaction among all households, one-person households, and households of the elderly. It reduced the housing cost burden for all household types. The status of receiving the housing allowance negatively affected the residential satisfaction for all households and increased the housing cost burden for young adult households. We present policy implications for future housing support programs based on the findings.

**Keywords:** housing support programs; residential satisfaction; housing cost burden; public rental housing program; housing allowance program



## 1. Introduction

Housing in locations with social and topographical advantages and favorable living environments generally has a high cost burden. In Korea, the pricing gap between housing with a favorable environment and that with an unfavorable environment is increasing, and the housing cost burden of households with a favorable housing environment is also increasing [1]. Those who cannot carry the cost burden are more likely to choose a housing area with a relatively poor housing environment, and this results in a low level of residential satisfaction. The issue of housing instability, manifested in residential dissatisfaction and a housing cost burden, significantly affects family finances as well as quality of life. Further, it negatively affects the community and the country. In particular, the housing problem can pose a serious financial hardship for low-income families, driving disadvantaged and vulnerable housing groups into a poor housing environment [2,3].

Policymakers strive to resolve the problem of housing instability by implementing various housing welfare policies. To ensure housing stability for low-income households, policy support is indispensable, and the government expects these policies to improve the housing environment and reduce the housing cost burden. Most countries have adequate housing standards for their citizens and implement various housing welfare policies to ensure a minimum level of housing [4]. In Korea, the government is implementing housing

support programs with the ultimate goal of housing stability for disadvantaged groups such as low-income families.

The common types of housing support programs include public rental housing programs, housing allowance programs, and housing loans for purchase or rental. Public rental housing is a common measure of the housing support program from the supplier's perspective, whereas housing allowances and housing loans for rental are typical measures from the user's perspective [5,6]. The public rental housing supply places a significant financial burden on the government, and the supply method is rigid. Due to these limitations, a consumer-subsidized housing support program has increasingly been implemented in recent years [7]. In particular, the housing allowance program is a representative example of the relatively new demand-side housing support program, which was implemented in 2015. Owing to the limitations in obtaining data that can be used to evaluate the policy's effect, sufficient research findings on the policy's effectiveness do not exist. In addition, the findings from previous studies are somewhat contradictory. Therefore, an empirical study on the policy effectiveness of housing support programs can be seen as timely.

In this study, we aimed to analyze empirically the effects of housing support programs in Korea on residential satisfaction and the housing cost burden. In particular, we examined the effects of two representative housing support programs—the public rental housing program and the housing allowance program—on residential satisfaction and the housing cost burden. To this end, we posed the following research questions: first, how does the status of residing in public rental housing and that of receiving a housing allowance affect residential satisfaction and the housing cost burden? Second, how does the effect of housing support programs on residential satisfaction and the housing cost burden differ based on household attributes?

Our paper is organized as follows: in Section 2, we have reviewed the current status of public rental housing and housing support programs in Korea, as well as previous studies on residential satisfaction and the burden of housing costs. Section 3 details the variables for empirical analysis we selected using data from the 2020 Housing Survey conducted by the Ministry of Land, Infrastructure, and Transport and provides explanations for the variables. Section 4 describes the multiple regression analysis and sequential logistic regression analysis we conducted to analyze how the housing support program affected residential satisfaction and the housing cost burden and how these effects change according to household characteristics. In Section 5, we have attempted to present policy implications for housing support programs based on the analysis results.

## 2. Literature Review

### 2.1. Representative Housing Support Programs: Public Rental Housing Program and Housing Allowance Program

To uphold human dignity and worth, a certain housing condition must be ensured, and a housing policy for low-income families, in particular, is important to guarantee the right to housing [4,8,9]. A housing support program aims to resolve the housing instability issue among low-income families and improve housing welfare [10,11]. As a housing support program for low-income households, the Korean government is operating housing support programs such as public rental housing, housing allowance programs, and housing loans for purchase or rental [12]. Policies related to housing support programs are categorized into supply-side and demand-side policies [13]. Public rental housing is a representative supply-side policy measure; housing support programs mainly comprise constructions of supply-side public rental housing [5,14]. Public rental housing is low-price rental housing that the government or public organizations provide to ensure a stable housing supply for low-income households. Different government administrations have called public rental housing by different names throughout the years, and since the supply of permanent rental housing began in 1989, various public rental housing has been supplied in earnest since 2008 [15]. In the early phase, the goal was to stabilize the housing price, but opinions on whether the intended effect was achieved are divided [16]. Eligible beneficiaries of the

public rental housing include household members with no housing, those with an average monthly income below 150% of the standard median income, those with total assets in the third income quintile, and those with average net assets below KRW 288 million (as of 2020). To provide low-income families with the opportunity to move into public rental housing, 60% of the housing supply is first offered to low-income families as follows: young adults, newlywed couples, and the elderly, accounting for 11%, 7%, and 10%, respectively.

However, it was pointed out that the quantitative-oriented supply-side policy did not take into account the quality of housing [17,18] and faced physical limitations, such as the lack of housing sites for supply. Owing to the limitations of implementing only the supply-side housing policy, the existing supply-side programs have been switched to a consumer-subsidized policy [4]. The housing allowance program is a representative demand-side housing support program, and the subject and scope of support are expanding. The housing allowance program is a system that subsidizes housing expenses for low-income families to stabilize their residence. The previous housing benefit was included in the integrated benefit, which was enforced in 2000 according to the National Basic Livelihood Security Act. The integrated benefit was provided to only those who were selected as recipients of the basic livelihood allowance, and these recipients were guaranteed to receive daily living subsidies, housing, medical care, and other subsidies. This clear-cut measure that supported all or none was limited in that the possibility that some disadvantaged groups would continue to live in poverty remained. In 2015, the integrated benefit system was changed to a customized support system, allowing even the non-recipients of basic livelihood security to receive the necessary support. Along with this reorganization, the supervising ministry was also changed from the Ministry of Health and Welfare to the Ministry of Land, Infrastructure and Transport. Renting families now receive the monthly rent subsidy, and homeowners receive a subsidy for home repair and maintenance of their housing facility. Ever since the payment system of the housing allowance program was reformed, the government has continued to increase the eligibility criteria and strived to provide a practical payment amount according to the level of the housing cost burden. Currently, households eligible for the housing allowance program include those with a recognized income below 46% of the standard median income. The government also pays differential amounts based on the recognized income, the number of household members, housing type, and housing cost burden. The amount of subsidy also differs based on the area of residence. To take one-person households as an example, the government pays KRW 327,000 in Seoul, KRW 253,000 in Gyeonggi and Incheon, and KRW 201,000 in other special case cities excluding six metropolitan cities, Sejong, and the Seoul metropolitan area (a metropolitan area comprising Seoul, Incheon, and Gyeonggi-do, located in northwest Korea. It is the residential, commercial, industrial, and cultural center of Korea). Additionally, housing benefit recipients can receive duplicate benefits from both housing benefits and the public rental housing system if their eligibility to move into public rental housing is recognized.

*2.2. Various Factors Affecting Residential Satisfaction and Housing Cost Burden*

Residential satisfaction is the difference between occupants' actual and expected residential attributes [19]. It depicts how satisfied a resident is with the current residence and housing environment, and this serves as an important reference for establishing a housing policy [20]. Residential satisfaction has been defined by focusing on various environmental factors along with the evaluation of housing needs' satisfaction. Residential satisfaction is an individual's subjective judgment that comprehensively considers the direct and indirect influencing factors of residential life, including the overall satisfaction with a house's physical facility elements and the surrounding environment [21,22].

Residential satisfaction is known to be affected by household attributes (such as age, gender, and income), housing attributes (such as housing type and occupancy type), and neighborhood attributes (such as accessibility to facilities and green space, public order, and neighbor attributes) [23–27]. Residential satisfaction is closely related to personal

variables, and factors such as income, age, gender, social status, family composition, and age are known to influence it [28]. Residential satisfaction is also known to influence the status of ownership, type of housing, and degree of social participation, as well as the surrounding environment; that is, the characteristics of the neighborhood environment near the residence [29,30]. For example, elderly households often have difficulty accessing community facilities that are important for daily life, which can affect physical and social health and welfare, which negatively affects residential satisfaction [31].

The housing cost burden refers to all expenses incurred continuously while living in a house; that is, the housing-related expenses for each household [32]. In Korea, the housing cost burden continues to grow due to the high housing and rent prices [33]. The housing cost burden is known to be affected by various factors such as household attributes and housing attributes. Among household attributes, sociodemographic factors (such as occupation and education level of the household head) and family factors (such as income and number of household members) affect the housing cost burden [34–36].

As for households' characteristics, sociodemographic factors, such as the household owner's occupation and educational background, and household factors, such as income and the number of household members, are known to influence the housing cost burden [34–36]. Along with educational background, the household owner's job is also known to influence the housing cost burden, and the low-income class is known to have a relatively high housing cost burden [33,37]. Further, the actual housing cost of a household varies depending on the household's residential characteristics and socioeconomic characteristics [35,38]. As for residential attributes, housing type, residential area, and residence period are known to influence the housing expense burden [32,39,40]. The degree of housing cost burden has been confirmed to vary depending on the type of house, region, and duration of residence, and the housing cost burden is significantly influenced in the case of monthly rental households [41]. In addition, a study on neighborhood attributes was conducted by expanding the range of factors affecting the housing cost burden, and accessibility to facilities such as public transportation was found to affect the housing cost burden [40].

### 2.3. The Effect of the Housing Support Programs on Residential Satisfaction and Housing Cost Burden

In urban space, the socially vulnerable or low-income class continuously experience instability in the residential environment [42]. The public sector needs to come up with various support measures to improve housing instability for these vulnerable groups. The policy goal of a housing support program is housing stability. Planners use housing support programs to help low-income households escape poverty and live in a better neighborhood [43]. Many countries have housing support programs, such as supplying affordable housing or providing vouchers, and these policies are showing some expected effects [44–46]. As a way of measuring whether the housing support program is achieving the policy purpose, we examined residential satisfaction and the housing cost burden. By so doing, we could verify the degree of improvement in housing instability.

Housing policy is one of the factors that influence residential satisfaction. Different types of housing policies have been found to exert their effects on residential satisfaction [47]. In particular, many researchers have studied how public rental housing, which is a representative supply-oriented housing support program, affects Korean people's residential satisfaction. They revealed that different types of public rental housing had different effects on residential satisfaction [25,48]. However, few studies have focused on how a demand-side housing support program affects residential satisfaction. Some studies have reported that the residential satisfaction of households receiving the housing allowance is lower than that of households residing in public rental housing. Contrarily, other studies have argued that the demand-side housing support program improved residential satisfaction to some extent [14,49].

Housing policy is one of the factors that influence the housing cost burden. Public rental housing, which is a representative supply-side policy, is generally known to reduce

tenants' housing cost burden [32,50]. In addition, due to the difference in the housing supply method for each housing type, the housing cost burden differs among various public rental housing types such as permanent rental housing and multi-household housing, purchased or rental housing [51]. Concerning the effect of the demand-side policy on housing cost burden, study results are conflicting. Some studies have reported that the customized housing allowance program revised in 2015 reduced renting households' housing cost burden. Other studies have reported that the households receiving the housing allowance had an increased housing cost burden compared to households residing in public rental housing or non-beneficiary households with a similar income level [7,37]. In addition, another finding suggested that the demand-side housing support program did not relieve the housing cost burden [49]. Aside from these, some scholars have also suggested that the housing allowance program as a driving force to improve housing standards was limited despite its contribution to reducing the rent and housing cost burden. As such, studies on the effect of the housing support program on the housing cost burden do not provide consistent findings [52].

*2.4. Limitations of Prior Studies*

As discussed above, many studies have focused on the effectiveness of the supply-side policy in Korea. However, few studies have analyzed the effectiveness of housing support programs that implemented the demand-side policy in the context of diversified housing support programs. In particular, few studies have empirically investigated the direct effects of housing support programs on residential satisfaction and the housing cost burden. Therefore, this study aimed to analyze the effectiveness of housing support programs and provide policy implications for efficient housing support programs in the future.

Prior studies on the effectiveness of housing support programs have mainly analyzed supply-side policies. In particular, most studies have empirically examined residential satisfaction and housing cost burden for public rental housing residents [48,51]. Studies on demand-side housing policies are relatively rare. Although studies on the effect of housing vouchers have been conducted in other countries [53,54], few Korean researchers have investigated the demand-side housing policy in Korea. Before the housing allowance program was revised, some researchers studied the residential satisfaction of households receiving the housing allowance and the effect of housing loans for *jeonsei* or purchase. Nonetheless, they failed to provide a thorough analysis due to a lack of available data about the new housing allowance program, which was implemented for only a short period [14,49,55]. Therefore, to derive more meaningful research results, we used the data from the 2020 Korea Housing Survey for analysis and data accrued up to five years since the implementation of the new housing allowance program.

Next, we divided the vulnerable housing groups based on household attributes and analyzed the effectiveness of the housing support programs. According to the Korean government's 2017 housing welfare roadmap, customized housing support plans for each stage of life and income level would be provided based on household composition (young adults, newlywed couples, the elderly, and low-income families) to implement the policy effectively. Further, more than 30% of one-person households were found to have an excessive housing cost burden, and this number has been rapidly increasing in recent years. This demonstrates the need for a housing policy that reflects the attributes of one-person households [56] as well. In this study, we analyzed the effect of the housing support programs based on household attributes by comprehensively considering the changes in the policy direction of the housing support programs.

## 3. Method

*3.1. Data and Conceptual Framework*

The study area was the entire country of Korea (see Figure 1). In this study, we used data from the 2020 Korea Housing Survey conducted by the Ministry of Land, Infrastructure and Transport. The total number of households and household heads in Korea at the time

of the survey made up the total survey population. The survey investigated overall matters of residential life, such as household characteristics, residential environment, and residential movement of the Korean people, to provide information necessary for policy establishment related to residential welfare improvement. The main items included housing characteristics, housing cost burden, residential satisfaction, and the status of home ownership of the current residential house. The structured survey was carried out from July to December 2020 throughout the country, via face-to-face interviews by trained interviewers. Rental households were extracted from the data of households in the Korea Housing Survey sample, and households that did not respond to the questionnaire related to public rental housing and housing benefits were excluded. Then, 2950 households were considered for the final analysis after removing outliers and missing values.

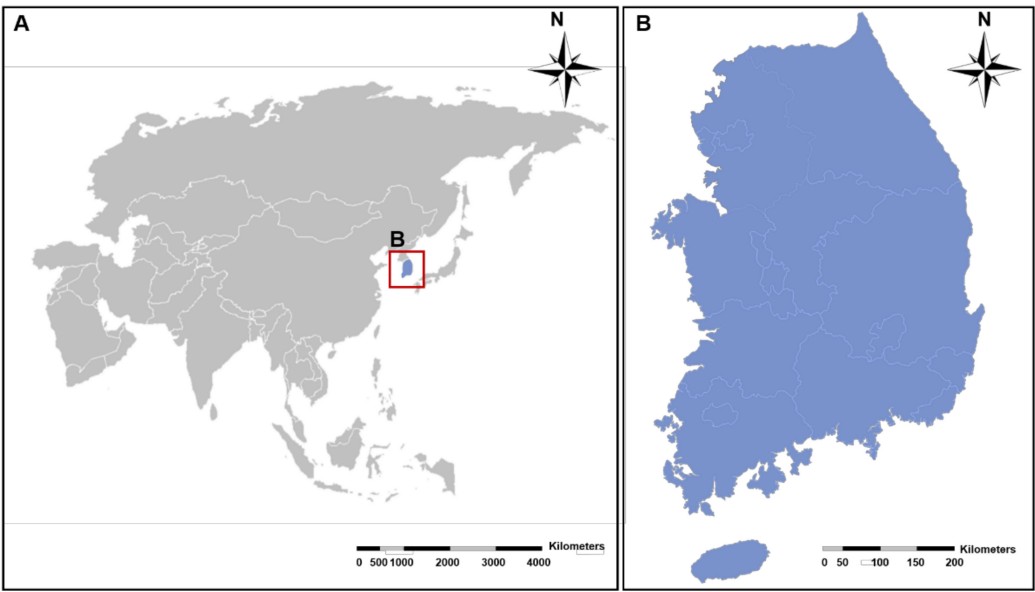

**Figure 1.** Study area: geographical location of (**A**) the continent of Asia, (**B**) the Republic of Korea.

For the classification of households, the criteria defined in Korea's housing policy were considered. One-person households are defined as households comprising one member regardless of the household owner's age. Newlyweds are defined as couples within seven years of the marriage registration period. In the case of young households, youth is defined as those under 39 years of age. In this study, young households are defined as households with a household owner under 39 years of age. Further, those 65 years of age or older comprise the elderly, and elderly households are households whose owners are 65 years of age or older. The final sample included 1504 surveys from one-person households, 189 surveys from newlywed couple households, 473 surveys from households of young adults, and 1089 surveys from households of the elderly.

We aimed to verify the effect of housing support programs on residential satisfaction and the housing cost burden. To this end, we conducted two analyses. In one, we considered "residential satisfaction" as the dependent variable and utilized multiple regression; in the other, we considered "housing cost burden" as the dependent variable and utilized ordinal logistic regression. The dependent variable of the housing satisfaction model was a continuous variable, and the dependent variable of the housing cost burden model was measured on a Likert scale. Therefore, we used multiple regression and ordered logistic regression analysis in consideration of the characteristics of the dependent variable. Figure 2 demonstrates the conceptual framework of this study.

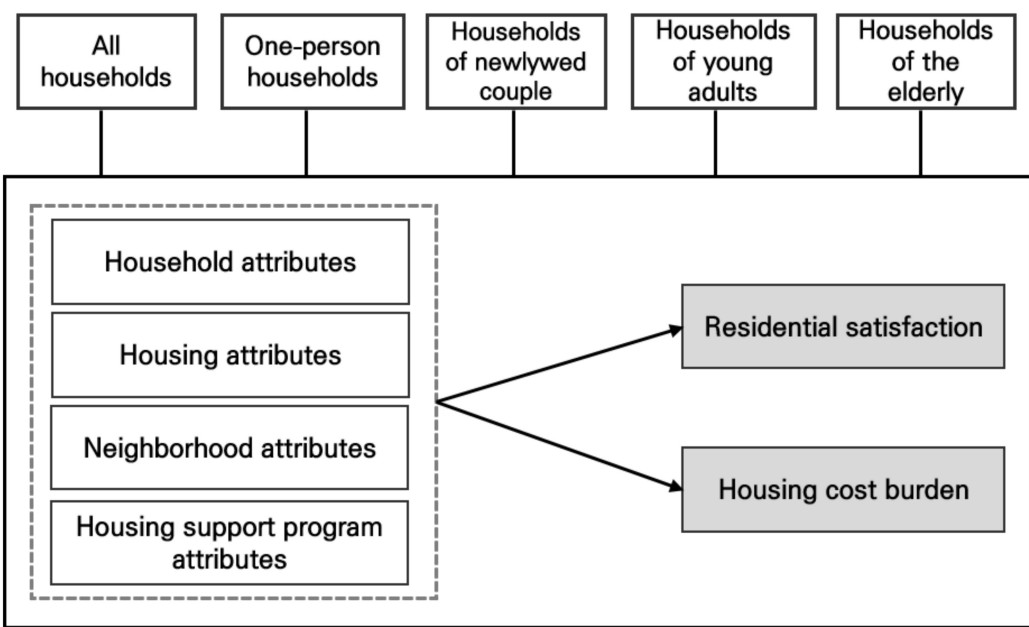

**Figure 2.** Conceptual framework.

### 3.2. Variables and Measurements

Table 1 presents the description of the dependent and independent variables. The dependent variables of this study are residential satisfaction and housing cost burden. The level of residential satisfaction is the average of the scores on the items related to "General satisfaction with housing condition" and "General satisfaction with neighborhood environment"; we measured these scores on a 4-point Likert scale. We used the level of housing cost burden as a variable after measuring it on a 4-point Likert scale in response to the degree of housing cost burden at the current residence. We implemented three methods to assess housing cost burden: ratio measurement method, residual income valuation method, and a behavioral (subjective) method [57]. The ratio measurement method, which is typically indicated as the rent-to-income ratio (RIR), calculates cash flow using rent as a measure. This method is limited in that it is difficult to accurately calculate the housing cost burden in the form of stock such as monthly rent [32,58]. The residual income valuation is a method of determining the affordability of housing by comparing the income after housing payment with the non-housing expenditure of the household. This method is also limited in that it is difficult to reflect the discrepancy in non-housing expenditures by region. In other words, this method is good for studying small areas [59]. This study aimed to analyze empirically whether the reduction of housing cost burden through the housing support program affects policy beneficiaries' subjective experience. In this regard, it must be mentioned that rental households in Korea pay housing costs not only in the form of cash but also often in the form of *jeonsei* and deposits. This means that the housing cost burden measurement method using RIR cannot reflect Korean rental households' actual housing cost burden [58]. Therefore, we considered a subjective judgment method as an appropriate method for measuring the housing cost burden. Further, we considered residents' subjective judgment regarding the current level of housing cost burden as a variable.

**Table 1.** Variable measurement.

| Variable | | Measurement |
|---|---|---|
| Dependent variable | Residential satisfaction | Average value of overall satisfaction with housing condition and neighborhood environment measured on a 4-point Likert scale. |
| | Housing cost burden | The degree to which the resident subjectively feels the housing cost burden of the current residential house: (1 = none, 2 = not much, 3 = some, 4 = very much). |
| Household attributes | Gender of household head | Male = 1, female = 0. |
| | Age of household head | Age of household head (years of age). |
| | Occupation of household head | Office job = 1, service/sales = 2, technical job = 3, other =4, unemployed = 5. |
| | Education level of household head | High school or below = 1, college = 2, graduate school or higher = 3. |
| | Low-income family | Household income below 50% of the median income = 1, other = 0. |
| | Number of household members | Number of household members living together (# number). |
| Independent variable — Housing attributes | Seoul metropolitan area | Residence in Seoul metropolitan area = 1, other areas = 0. |
| | Occupancy period | Period of occupying the current housing (years). |
| | Housing type | Types of current residence: private housing = 1, apartment = 2, townhouse/multi-family housing = 3, studio = 4, other = 5. |
| | Monthly rental house | The occupancy type of the current housing: monthly rental house = 1, other = 0. |
| Neighborhood attributes | Accessibility to other facilities | The average score of questions related to satisfaction regarding the accessibility to four types of facilities (commercial facilities, medical facilities, cultural facilities, and public institutions) measured on a 4-point Likert scale. |
| | Accessibility to parks and green | Satisfaction with the accessibility to city parks and green space (4-point Likert scale). |
| | Accessibility to public transportation | Satisfaction with the accessibility to public transportation (4-point Likert scale). |
| | Public order | Satisfaction with the level of public order and crime prevention (4-point Likert scale). |
| | Relationship with neighbors | Satisfaction with the relationship with neighbors (4-point Likert scale). |

**Table 1.** *Cont.*

| Variable | Measurement | |
|---|---|---|
| | Public rental housing | User status of public rental housing: currently using = 1, currently not using = 0. |
| Housing support program attributes | Housing allowance program | User status of rental assistance as a housing allowance program: currently using = 1, currently not using = 0. |

Based on the review of previous studies, we selected the following as independent variables: household attributes, housing attributes, neighborhood environment attributes, and housing support programs' attributes. As variables of household attributes, we selected the following: gender, age, occupation, and education level of the household head, as well as low-income class status and the number of household members. Not all beneficiaries of the housing support program are low-income families because the selection criteria are different for each type of housing support program. Conversely, not all low-income people are automatically beneficiaries of the housing support program, so we considered low-income people as control variables for the residential satisfaction and housing cost burden model. We classified low-income households as households with a median income of 50% or less of the median income according to the criteria for median income by the number of household members. This standard is used for selecting the recipients of various welfare projects, including Korea's basic living security system. In the survey, we directly asked participants about their monthly income, but we divided the income variable, which is a continuous variable, into categories and used them to determine the difference in the effectiveness of housing support programs for low-income and non-low-income households. As variables of housing attributes, we selected the following: residence in the Seoul Metropolitan Area, occupancy period, housing type, and occupancy type. We classified housing types into private housing, apartments, townhouse/multi-family housing, studio, and others, and all these housing types could be paid for via monthly rent or considered ownership housing. As variables of neighborhood attributes, we selected the following: accessibility to facilities, parks and green space accessibility, public transportation accessibility, public order, and relationships with neighbors. As variables of housing support programs, we selected the status of residing in public rental housing and the status of receiving rental assistance such as a housing allowance. Public rental housing and the housing allowance can be considered typical examples of a supply-side housing policy and a demand-side housing policy, respectively. We selected them as variables because they are the most significant as supply-side and demand-side housing policies, respectively, thereby making it easy to obtain significant results through analysis.

## 4. Results

### 4.1. Descriptive Statistics

Table 2 presents the descriptive statistics of variables used in the analysis. As for the whole sample, the mean of each dependent variable—residential satisfaction and housing cost burden—was 2.917 (sd = 0.46) and 2.923 (sd = 0.51), respectively. When we examined the dependent variables based on household attributes, the one-person household group had the lowest level of residential satisfaction and the highest housing cost burden among all household types. The newlywed couple household group had the highest level of residential satisfaction, and the young adult household group had the lowest level of housing cost burden. Of all the data used for the analysis, the descriptive statistics of the whole sample are as follows. Low-income families accounted for 50%, and the average number of household members was 1.8 (sd = 1.13). Households residing in the Seoul metropolitan area accounted for 39.7%, and the average occupancy period was

6.1 years (sd = 6.04). In terms of housing type, apartments had the highest percentage at 48.41%. Further, 87.2% of all households paid monthly rent. In terms of satisfaction with the neighborhood, the mean for each variable was as follows: 2.847 (sd = 0.54) for accessibility to other facilities, 2.951 (sd = 0.71) for accessibility to parks and green space, 3.033 (sd = 0.63) for public transportation accessibility, 3.016 (sd = 0.55) for public order, and 2.988 (sd = 0.51) for the relationship with neighbors.

**Table 2.** Descriptive statistics.

| | | All Households | One-Person Households | Household of Newlywed Couples | Households of Young Adults | Households of the Elderly |
|---|---|---|---|---|---|---|
| | | Mean/Frequency (S.D./%) | Mean/Frequency (S.D./%) | Mean/Frequency (S.D./%) | Mean/Frequency (S.D./%) | Mean/Frequency (S.D./%) |
| Dependent variable | residential satisfaction | 2.917 (0.46) | 2.900 (0.51) | 2.979 (0.53) | 2.970 (0.55) | 2.912 (0.48) |
| | Housing cost burden | 2.923 (0.51) | 2.921 (0.83) | 2.884 (0.77) | 2.827 (0.79) | 2.901 (0.82) |
| Household attributes | Gender of household head | 0.621 (0.49) | 0.503 (0.50) | 0.989 (0.10) | 0.721 (0.45) | 0.512 (0.50) |
| | Age of household head | 57.583 (16.16) | 61.500 (16.59) | 36.386 (5.22) | 32.677 (4.61) | 74.502 (6.96) |
| | Occupation of household head — Office job | 395 (13.39) | 132 (8.78) | 100 (52.91) | 219 (46.30) | 3 (0.28) |
| | Service/sales | 441 (14.95) | 60 (10.64) | 38 (20.11) | 111 (23.47) | 31 (2.85) |
| | Technical job | 284 (9.63) | 86 (5.72) | 34 (17.99) | 59 (12.47) | 31 (2.85) |
| | Other | 280 (9.49) | 110 (7.31) | 10 (5.29) | 17 (3.59) | 109 (10.01) |
| | Unemployed | 1550 (52.54) | 1016 (67.55) | 7 (3.70) | 67 (14.16) | 915 (84.02) |
| | Education level of household head — High school | 2247 (76.17) | 1267 (84.24) | 45 (23.81) | 158 (33.40) | 1057 (97.06) |
| | College | 663 (22.47) | 226 (15.03) | 134 (70.90) | 302 (63.85) | 28 (2.57) |
| | Graduate school | 40 (1.36) | 11 (0.73) | 10 (5.29) | 13 (2.75) | 4 (0.37) |
| | Low-income families | 0.500 (0.500) | 0.632 (0.482) | 0.05 (0.214) | 0.146 (0.353) | 0.796 (0.403) |
| | Number of household members | 1.885 (1.13) | 1.000 (0.00) | 2.815 (0.78) | 2.053 (1.14) | 1.418 (0.71) |
| Housing attributes | Seoul metropolitan areas | 0.397 (0.49) | 0.408 (0.49) | 0.540 (0.50) | 0.488 (0.50) | 0.379 (0.49) |
| | Occupancy period | 6.129 (6.04) | 6.217 (6.32) | 2.656 (1.53) | 2.579 (1.72) | 8.364 (7.41) |
| | Housing type — Private housing | 969 (32.85) | 588 (39.10) | 28 (14.81) | 138 (29.18) | 380 (34.89) |
| | Apartment | 1428 (48.41) | 570 (37.90) | 123 (65.08) | 203 (42.92) | 560 (51.42) |
| | Townhouse/ multi-family housing | 314 (10.64) | 132 (8.78) | 33 (17.46) | 64 (13.53) | 86 (7.90) |
| | Studio | 83 (2.81) | 68 (4.52) | 5 (2.65) | 50 (10.57) | 6 (0.55) |
| | Other | 156 (5.29) | 146 (9.71) | - | 18 (3.81) | 57 (5.23) |
| | Monthly rental house | 0.872 (0.33) | 0.936 (0.24) | 0.497 (0.50) | 0.672 (0.47) | 0.961 (0.19) |
| Neighborhood attributes | Accessibility to other facilities | 2.847 (0.54) | 2.817 (0.55) | 2.968 (0.55) | 2.953 (0.55) | 2.779 (0.54) |
| | Accessibility to parks/green space | 2.951 (0.71) | 2.882 (0.72) | 3.016 (0.60) | 2.915 (0.70) | 2.958 (0.71) |
| | Accessibility to public transportation | 3.033 (0.63) | 3.048 (0.63) | 3.026 (0.65) | 3.076 (0.66) | 3.026 (0.62) |
| | Public order | 3.016 (0.55) | 2.981 (0.55) | 3.090 (0.54) | 3.044 (0.58) | 3.018 (0.54) |
| | Relationship with neighbors | 2.988 (0.51) | 2.951 (0.52) | 3.000 (0.48) | 2.956 (0.51) | 3.028 (0.49) |
| Housing support program attributes | Public rental housing | 0.428 (0.49) | 0.408 (0.49) | 0.265 (0.44) | 0.207 (0.41) | 0.562 (0.50) |
| | Housing allowance program | 0.487 (0.50) | 0.612 (0.49) | 0.048 (0.21) | 0.121 (0.33) | 0.750 (0.43) |
| | N | 2950 | 1504 | 189 | 473 | 1089 |

Households residing in public rental housing accounted for 42.8%, and households receiving a housing allowance accounted for 48.7%. The households of the elderly (56.2%) had the highest rate of occupancy in public rental housing, whereas young adult households had the lowest rate of occupancy in public rental housing (20.7%). Further, the households of the elderly (75%) had the highest rate of receiving a housing allowance,

whereas the newlywed couple household group had the lowest rate of receiving a housing allowance (4.8%).

### 4.2. Factors Influencing Residential Satisfaction

Table 3 presents the regression analysis results regarding residential satisfaction. The findings for the whole sample were as follows. Among household attributes, gender, occupation, education level, and the number of household members significantly affected residential satisfaction. In terms of gender, households with a male household head had a higher level of residential satisfaction. In terms of occupation, households whose heads had office jobs had a higher level of residential satisfaction than those whose heads had technical jobs. In terms of the household head's education level, the level of residential satisfaction was lower among households whose heads had college degrees than those whose heads had high school degrees. Further, the number of household members negatively affected residential satisfaction. Of all housing attributes, occupancy period and housing type significantly affected residential satisfaction. In particular, occupancy period negatively affected residential satisfaction. Among the variables of housing types, households residing in apartments, townhouse/multi-family housing, or studio had a higher level of residential satisfaction than those residing in private housing. Among neighborhood attributes, accessibility to other facilities, accessibility to parks and greens, public transportation accessibility, public order, and the relationship with neighbors all had a significant positive effect on residential satisfaction. Of all the housing support program attributes, the variables of public rental housing and the housing allowance program showed statistically significant effects. In particular, households residing in public rental housing had a higher level of residential satisfaction than those residing in other types of housing. Households receiving a housing allowance had lower residential satisfaction than those not receiving a housing allowance.

**Table 3.** Results of residential satisfaction model.

| | | | Whole Sample | One-Person Households | Households of Newlywed Couples | Households of Young Adults | Households of the Elderly |
|---|---|---|---|---|---|---|---|
| | | | Coef. (S.E.) | Coef. (S.E.) | Coef. (S.E.) | Coef. (S.E.) | Coef. (S.E.) |
| Constant | | | 0.847 * (0.06) | 0.739 *** (0.10) | 0.735 * (0.41) | 0.765 *** (0.17) | 1.122 *** (0.25) |
| Household attributes | Gender of household head | | 0.621 (0.49) | 0.503 (0.50) | 0.989 (0.10) | 0.721 (0.45) | 0.512 (0.50) |
| | Age of household head | | 0.007 ** (0.01) | 0.000 (0.02) | 0.165 (0.27) | −0.026 (0.04) | 0.025 (0.02) |
| | Occupation of household head (ref. office) | Service/sales | −0.028 (0.03) | 0.023 (0.05) | −0.040 (0.08) | 0.023 (0.04) | −0.254 (0.21) |
| | | Technical job | −0.076 ** (0.03) | −0.016 ** (0.05) | −0.072 (0.09) | −0.107 ** (0.05) | −0.346 * (0.21) |
| | | Other | −0.055 (0.03) | −0.076 (0.05) | 0.102 (0.15) | −0.022 (0.09) | −0.295 (0.20) |
| | | Unemployed | −0.027 (0.03) | 0.000 (0.05) | −0.341 (0.21) | −0.029 (0.06) | −0.274 (0.20) |
| | Education level of household head (ref. high school) | College | −0.034 * (0.02) | 0.033 (0.03) | −0.083 (0.08) | −0.067 (0.04) | −0.068 (0.07) |
| | | Graduate school | −0.072 (0.06) | −0.028 (0.11) | −0.141 (0.15) | −0.064 (0.10) | 0.034 (0.17) |
| | Low-income families | | −0.003 (0.02) | −0.005 (0.03) | 0.085 (0.21) | −0.055 (0.06) | 0.037 (0.03) |
| | Number of household members | | −0.018 *** (0.01) | - | −0.025 (0.04) | −0.045 *** (0.02) | −0.005 (0.02) |

**Table 3.** *Cont.*

| | | Whole Sample | One-Person Households | Households of Newlywed Couples | Households of Young Adults | Households of the Elderly |
|---|---|---|---|---|---|---|
| | | Coef. (S.E.) | Coef. (S.E.) | Coef. (S.E.) | Coef. (S.E.) | Coef. (S.E.) |
| Housing attributes | Seoul metropolitan areas | −0.006 (0.01) | −0.015 (0.02) | −0.009 (0.06) | −0.077 ** (0.03) | 0.031 (0.02) |
| | Occupancy period | −0.004 *** (0.00) | −0.003 ** (0.00) | 0.009 (0.02) | −0.001 (0.01) | −0.004 ** (0.00) |
| | Housing type (ref. private housing) — Apartment | 0.069 *** (0.02) | 0.062 ** (0.03) | −0.036 (0.09) | −0.003 (0.04) | 0.132 *** (0.03) |
| | Townhouse/ multi-family housing | 83 (2.81) | 68 (4.52) | 5 (2.65) | 50 (10.57) | 6 (0.55) |
| | Studio | 0.125 *** (0.04) | 0.086 * (0.05) | 0.088 (0.20) | 0.117 ** (0.06) | 0.093 (0.14) |
| | Other | −0.142 ***(0.03) | −0.136 *** (0.03) | - | −0.028 (0.08) | −0.227 *** (0.05) |
| | Monthly rental house | 0.021 (0.02) | 0.023 (0.04) | −0.014 (0.07) | −0.020 (0.04) | 0.019 (0.05) |
| Neighborhood attributes | Accessibility to other facilities | 0.213 *** (0.01) | 0.229 *** (0.02) | 0.188 ** (0.07) | 0.257 *** (0.04) | 0.176 *** (0.02) |
| | Accessibility to parks/greens | 0.072 *** (0.01) | 0.080 *** (0.01) | 0.173 *** (0.05) | 0.107 *** (0.03) | 0.072 *** (0.02) |
| | Accessibility to public transportation | 0.038 *** (0.01) | 0.045 *** (0.02) | 0.008 (0.06) | 0.004 (0.03) | 0.078 *** (0.02) |
| | Public order | 0.223 *** (0.01) | 0.239 *** (0.02) | 0.258 *** (0.06) | 0.233 *** (0.03) | 0.180 *** (0.02) |
| | Relationship with neighbors | 0.184 *** (0.01) | 0.164 *** (0.02) | 0.173 ** (0.07) | 0.202 *** (0.04) | 0.167 *** (0.02) |
| Housing support Program attributes | Public rental housing | 0.092 *** (0.02) | 0.110 *** (0.03) | 0.056 (0.08) | 0.057 (0.05) | 0.061 * (0.03) |
| | Housing allowance program | −0.034 * (0.02) | −0.028 (0.03) | −0.031 (0.18) | 0.030 (0.07) | −0.032 (0.03) |
| | N | 2950 | 1504 | 189 | 473 | 1089 |
| | F | 106.85 *** | 61.09 *** | 6.58 *** | 25.71 *** | 36.35 *** |
| | Log likelihood | −966.762 | −503.004 | −59.894 | −118.274 | −330.372 |
| | $R^2$ | 0.467 | 0.487 | 0.478 | 0.579 | 0.451 |
| | Adjust $R^2$ | 0.463 | 0.479 | 0.406 | 0.557 | 0.438 |

In the one-person household group, households with one member are the subject of analysis, so the variable 'number of members' is excluded from the analysis. *** $p < 0.01$, ** $p < 0.05$, * $p < 0.1$.

The findings for the one-person household group were as follows. Among household attributes, the age and occupation of the household head significantly influenced residential satisfaction. In particular, the older the household head, the lower the residential satisfaction. In terms of occupation, those with office jobs had higher residential satisfaction than those with technical jobs. Among housing attributes, occupancy period and housing type significantly affected residential satisfaction. In particular, the occupancy period negatively affected residential satisfaction. Among the variables of housing types, households residing in an apartment or studio had a higher level of residential satisfaction than those residing in private housing. Among neighborhood attributes, accessibility to other facilities, accessibility to parks and greens, public transportation accessibility, public order, and the relationship with neighbors had significant positive effects on residential satisfaction. Among the variables of housing support programs, only public rental housing had a significant effect; and households residing in public rental housing had a higher level of residential satisfaction than those not residing in public rental housing.

The findings for the newlywed couple household group were as follows. None of the household and housing attributes significantly affected residential satisfaction. Among neighborhood attributes, accessibility to other facilities, accessibility to parks and greens, public order, and the relationship with neighbors positively affected residential satisfaction. None of the housing support program variables significantly affected residential satisfaction.

The findings for the young adult household group were as follows. Among household attributes, the occupation of the household head and the number of household members significantly affected residential satisfaction. Households whose household heads had office jobs had a higher level of residential satisfaction than those whose household heads had technical jobs. Further, the number of household members negatively affected residential satisfaction. Among housing attributes, housing type and residence in the Seoul metropolitan area significantly affected residential satisfaction. In particular, households residing in the Seoul metropolitan area had a lower level of residential satisfaction than those residing in other areas. As for young adult households, the percentage of employment of the household head was higher than that of other household groups. This can be seen as the representation of the reality that households living in the Seoul metropolitan area due to work cannot afford high housing costs in areas with favorable housing conditions. Among the variables of housing types, households residing in townhouse/multi-family housing or studio had a higher level of residential satisfaction than those residing in private housing. Among neighborhood attributes, accessibility to other facilities, accessibility to parks/greens, public order, and the relationship with neighbors positively affected residential satisfaction. Finally, the user status of public rental housing and the housing allowance program did not have a significant effect on residential satisfaction.

The findings for the households of the elderly were as follows. Among household attributes, the occupation of the household head was the only variable that significantly affected residential satisfaction. In particular, households whose household heads had office jobs had a higher level of residential satisfaction than those whose household heads had technical jobs. Of all housing attributes, occupancy period and housing type significantly affected residential satisfaction. Occupancy period negatively affected residential satisfaction. Among the variables of housing types, households residing in an apartment had a higher level of residential satisfaction than those residing in private housing. Among neighborhood attributes, accessibility to other facilities, accessibility to parks and greens, public transportation accessibility, public order, and the relationship with neighbors positively affected residential satisfaction. Among the attributes of housing support programs, households residing in public rental housing had a higher level of residential satisfaction than those who did not reside in public rental housing.

Our findings regarding the variables that significantly affected residential satisfaction, such as household, housing, and housing environment attributes, are consistent with previous findings [27,47]. Among housing characteristics, the period of residence negatively affected residential satisfaction in all households, one-person households, and elderly households. This may have led to a decrease in residential satisfaction as residence period increased. Most housing environment variables affected residential satisfaction irrespective of household attributes.

The user status of public rental housing positively affected residential satisfaction among all households, one-person households, and households of the elderly. This means that public rental housing increased residential satisfaction among households that did not have their own home.

In contrast, public rental housing did not have a significant effect on residential satisfaction among the newlywed couple and young adult household groups. This may be due to the low user status of these two household groups. The housing allowance program had a significant positive effect on the residential satisfaction of only the whole sample group. Its effect on other household groups was negative. This may be because of their area of residence. Most households benefiting from the housing allowance program resided in areas with a poor housing environment and were from low-income families.

### 4.3. Factors Influencing Housing Cost Burden

Table 4 presents the results of the logical regression analysis related to the housing cost burden. The pseudo R2 value of the housing cost burden model was relatively lower than the R2 value of the housing satisfaction model, but in general, the pseudo R2 value tends to be significantly lower than R2 in OLS [60]. In addition, this analysis aimed to find out how the independent variable affects the housing cost burden, and it was judged that the low pseudo R2 value was not a big problem. The findings for whole sample were as follows. Among household attributes, the occupation and education level of the household head, as well as the number of household members significantly affected the housing cost burden. The housing cost burden was greater for households whose heads had office jobs than those whose heads had technical jobs. The housing cost burden was also greater for households whose heads had other jobs or no jobs at all than those whose heads had office jobs. The number of household members positively affected the housing cost burden. Among housing attributes, residence in Seoul metropolitan areas, occupancy period, housing type, and monthly rent significantly affected the housing cost burden. Households residing in the Seoul metropolitan area were more likely to have a greater housing cost burden than those residing in other areas. The occupancy period negatively affected the housing cost burden. Among the variables of housing types, households residing in an apartment were more likely to have a greater housing cost burden than those residing in private housing. Among the variables of occupancy types, households paying monthly rent were more likely to have a greater housing cost burden than those not paying monthly rent. Among neighborhood attributes, the relationship with neighbors negatively affected the housing cost burden. Among the housing support program variables, households residing in public rental housing were more likely to have a low housing cost burden than households not residing in public rental housing.

**Table 4.** Results of housing cost burden model.

| | | Whole Sample | | One-Person Households | Households of Newlywed Couples | Households of Young Adults | Households of the Elderly |
|---|---|---|---|---|---|---|---|
| | | Coef. (S.E.) | | Coef. (S.E.) | Coef. (S.E.) | Coef. (S.E.) | Coef. (S.E.) |
| Household attributes | Gender of household head | 0.911 (0.073) | | 0.957 (0.103) | 0.000 (0.000) | 0.715 (0.165) | 0.780 * (0.100) |
| | Age of household head | 1.002 (0.003) | | 1.002 (0.004) | 1.021 (0.033) | 1.020 (0.024) | 0.997 (0.009) |
| | Occupation of household head (ref. office) | Service/sales | 1.166 (0.173) | 1.248 (0.314) | 0.830 (0.360) | 1.080 (0.272) | 0.679 (0.715) |
| | | Technical job | 0.720 * (0.121) | 0.559 * (0.166) | 0.505 (0.240) | 0.709 (0.218) | 0.345 (0.368) |
| | | Other | 1.716 *** (0.315) | 1.779 * (0.536) | 1.939 (1.691) | 1.378 (0.734) | 1.250 (1.279) |
| | | Unemployed | 1.528 ** (0.262) | 1.685 * (0.480) | 1.059 (1.187) | 1.390 (0.538) | 1.101 (1.118) |
| | Education level of household head (ref. high school) | College | 0.765 ** (0.088) | 0.790 (0.147) | 0.688 (0.329) | 1.219 (0.302) | 0.731 (0.270) |
| | | Graduate school | 0.724 (0.238) | 0.843 (0.538) | 0.392 (0.336) | 0.384 (0.236) | 0.421 (0.451) |
| | Low-income families | 1.024 (0.114) | | 0.991 (0.164) | 0.460 (0.549) | 0.530 * (0.201) | 0.837 (0.149) |
| | Number of household members | 1.182 *** (0.043) | | - | 1.887***(0.409) | 1.276**(0.127) | 1.258**(0.115) |

**Table 4.** *Cont.*

| | | Whole Sample | One-Person Households | Households of Newlywed Couples | Households of Young Adults | Households of the Elderly |
|---|---|---|---|---|---|---|
| | | Coef. (S.E.) | Coef. (S.E.) | Coef. (S.E.) | Coef. (S.E.) | Coef. (S.E.) |
| Housing attributes | Seoul metropolitan areas | 1.226 *** (0.095) | 1.177 (0.126) | 2.255 ** (0.817) | 2.486 *** (0.511) | 1.363 ** (0.172) |
| | Occupancy period | 0.987 ** (0.006) | 0.993 (0.008) | 0.844 (0.094) | 0.934 (0.052) | 0.997 (0.008) |
| | Apartment | 1.604 *** (0.169) | 1.818 *** (0.292) | 1.968 (1.017) | 1.693 ** (0.442) | 1.608 ** (0.314) |
| Housing type(ref. private housing) | Townhouse/multi-family housing | 0.946 (0.125) | 1.289 (0.247) | 0.794 (0.470) | 0.844 (0.259) | 0.981 (0.236) |
| | Studio | 0.860 (0.205) | 0.953 (0.256) | 0.119 * (0.151) | 0.527 * (0.186) | 2.423 (2.229) |
| | Other | 1.301 (0.227) | 1.408 * (0.265) | - | 0.889 (0.478) | 1.259 (0.358) |
| | Occupancy type | 2.209 *** (0.265) | 2.135 *** (0.449) | 3.628 *** (1.513) | 2.031 *** (0.459) | 4.932 *** (1.478) |
| Neighborhood attributes | Accessibility to other facilities | 0.972 (0.079) | 0.973 (0.108) | 0.690 (0.285) | 0.655 * (0.151) | 0.946 (0.120) |
| | Accessibility to parks/greens | 0.946 (0.054) | 0.972 (0.076) | 1.309 (0.372) | 0.902 (0.139) | 0.910 (0.084) |
| | Accessibility to public transportation | 1.095 (0.072) | 1.026 (0.095) | 1.794 * (0.561) | 1.334 * (0.232) | 0.993 (0.106) |
| | Public order | 0.926 (0.071) | 0.981 (0.104) | 0.786 (0.281) | 1.061 (0.215) | 0.932 (0.119) |
| | Relationship with neighbors | 0.701 *** (0.056) | 0.656 *** (0.071) | 0.426 ** (0.169) | 0.572 ** (0.125) | 0.713 ** (0.099) |
| Housing support Program attributes | Public rental housing | 0.239 *** (0.025) | 0.212 *** (0.034) | 0.132 *** (0.062) | 0.249 *** (0.070) | 0.210 *** (0.040) |
| | Housing allowance program | 0.970 (0.102) | 1.011 (0.152) | 1.304 (1.264) | 3.556 *** (1.459) | 0.869 (0.134) |
| | /Cut1 | −3.596 (0.379) | −3.604 (0.548) | −21.231 (1137.494) | −3.537 (1.064) | −4.029 (1.345) |
| | /Cut2 | −1.502 (0.370) | −1.619 (0.538) | −18.544 (1137.494) | −1.435 (1.049) | −1.928 (1.339) |
| | /Cut3 | 0.929 (0.370) | 0.665 (0.538) | −15.604 (1137.494) | 1.210 (1.050) | 0.450 (1.338) |
| | N | 2950 | 1504 | 189 | 473 | 1089 |
| | F | 367.70 *** | 203.83 *** | 67.48 *** | 82.35 *** | 177.78 *** |
| | Log likelihood | −3240.008 | −1694.800 | −181.317 | −509.344 | −1203.468 |
| | Pseudo $R^2$ | 0.130 | 0.140 | 0.335 | 0.177 | 0.166 |

In the one-person household group, households with one member are the subject of analysis, so the variable 'number of members' is excluded from the analysis. *** $p < 0.01$, ** $p < 0.05$, * $p < 0.1$.

The findings for one-person households were as follows. Among household attributes, only the occupation of the household head significantly affected the housing cost burden. Households whose heads had office jobs had a greater housing cost burden than those whose heads had technical jobs. Further, household heads with other jobs or no jobs were more likely to have a greater housing cost burden than household heads with office jobs. Among housing attributes, housing type and monthly rent significantly increased the housing cost burden. Those who resided in an apartment were more likely to have a greater housing cost burden than those who resided in private housing. Households paying monthly rent were more likely to have a greater housing cost burden than other households. Among neighborhood attributes, the relationship with neighbors negatively affected the housing cost burden. Among the housing support program variables, households residing

in public rental housing were more likely to have a low housing cost burden than those residing in public rental housing.

The findings for the newlywed couple household group were as follows. Among household attributes, the number of household members positively affected the housing cost burden. Among housing attributes, housing type and monthly rent significantly affected the housing cost burden. Among the variables of housing types, households that resided in an apartment had a greater housing cost burden than those that resided in private housing. Households that paid monthly rent had a greater housing cost burden than those that did not pay monthly rent. Among neighborhood attributes, public transportation accessibility and the relationship with neighbors significantly affected the housing cost burden. The higher the satisfaction with public transportation accessibility, the higher the housing cost burden, which is generally considered to be the result of higher housing costs located in places with good public transportation accessibility. The relationship with neighbors negatively affected the housing cost burden. Among housing support program attributes, households residing in public rental housing were more likely to have a low housing cost burden compared to those not residing in public rental housing.

Next, the findings on young adult households were as follows. Among household attributes, the number of household members positively affected the housing cost burden. Among housing attributes, residence in the Seoul metropolitan area, housing type, and monthly rent significantly affected the housing cost burden. Households residing in the Seoul metropolitan area were more likely to have a greater housing cost burden than those residing in other areas. Among the variables of housing types, households residing in an apartment were more likely to have a greater housing cost burden than those residing in private housing. Further, households residing in private housing had a greater housing cost burden than those residing in studios. Households paying monthly rent were more likely to have a greater housing cost burden than those not paying a monthly rent. Among neighborhood attributes, accessibility to other facilities and public transportation accessibility, and the relationship with neighbors significantly affected the housing cost burden. Accessibility to facilities and relationship with neighbors negatively affected the housing cost burden, whereas public transportation accessibility positively affected the housing cost burden. Among housing support program attributes, both public rental housing and the housing allowance program, which were dummy variables, significantly affected the housing cost burden. Households residing in public rental housing were more likely to have a low housing cost burden than those not residing in public rental housing. Households using a housing allowance program were more likely to have a high housing cost burden than those not using a housing allowance program.

The findings for the households of the elderly were as follows. Among household attributes, the gender of the household head and the number of household members significantly affected the housing cost burden. In particular, households with a female household head were more likely to have a high housing cost burden than those with a male household head. The number of household members positively affected the housing cost burden. Among housing attributes, residence in the Seoul metropolitan area, housing type, and monthly rent significantly affected the housing cost burden. In particular, households residing in the Seoul metropolitan area were more likely to have a high housing cost burden than those living in other areas. Among the variables of housing types, households residing in an apartment were more likely to have a high housing cost burden than those residing in private housing. Households paying monthly rent were more likely to have a high housing cost burden than those not paying monthly rent. Among neighborhood attributes, the relationship with neighbors negatively affected the housing cost burden. Among the housing support program attributes, households residing in public rental housing were more likely to have a low housing cost burden than those not residing in public rental housing.

To sum up, the analysis of housing cost burden in all household groups showed that households paying monthly rent had a greater housing cost burden than those not

paying a monthly rent. In all household groups, except for one-person households, those who resided in the Seoul metropolitan area had a greater housing cost burden than those who did not. This finding is in line with that of previous research that households paying monthly rent and households residing in the Seoul metropolitan area have a greater housing cost burden than other households [7,32,61]. Further, public rental housing decreased the housing cost burden for all household groups. This indicates that public rental housing has a lower housing cost burden than other types of rental housing [62]. Among young adult households, households that received a housing allowance had a greater housing cost burden than those that did not receive it. Finally, the benefit amount of housing assistance is insufficient to reduce the housing cost burden for households eligible for the housing allowance program.

## 5. Discussion

Housing support programs are public policies to ensure minimum housing rights, and research has been steadily conducted on how various housing support programs affect housing stability. This study confirmed the effect on housing satisfaction and the housing cost burden to evaluate the effect of the housing support program in Korea. As widely known from previous studies, it was confirmed that the supply-oriented policy had a positive effect on housing satisfaction and reduction of the housing cost burden. However, the demand-oriented policy was not found to have a significant effect on housing satisfaction and reduction of the housing cost burden, as known from previous studies.

Based on our findings, we provide the following policy suggestions for the housing support programs. First, to maximize the effectiveness of the policy when screening eligible beneficiaries of the public rental housing program and the housing allowance program, the ratio must be considered based on household attributes. As for the public rental housing policy, it had a positive effect on reducing the housing cost burden. However, it is not fiscally or physically feasible to keep increasing the supply of public rental housing. When an integrated public rental housing policy is established in the future, the policy's effectiveness must be maximized by properly allocating the preferential supply ratio according to household attributes. As for the housing allowance program, support must be increased for those households that urgently need housing cost assistance. Among the households residing in rental housing, one-person households and elderly households had a higher monthly rent as well as a greater cost burden than other household groups. Therefore, the government must increase support for those households that urgently need housing assistance to increase the effectiveness of the housing allowance program.

Second, to increase the effectiveness of the housing allowance program, the current eligibility criteria and payment must be reviewed. According to a study conducted in an early stage when the housing allowance program was revised and implemented, backup measures for the program operation were needed because the housing assistance program did not have a significant effect [14,55]. Since the program's revision in 2015 to ensure that it is tailored to beneficiaries' needs, the Korean government has worked towards increasing the eligibility of the housing allowance program and offered practical housing assistance based on the level of housing cost burden. Continuing to make such improvements will help increase the program's effectiveness. Further, considering that the renting households in the Seoul metropolitan area carry a relatively high housing cost burden, a comprehensive and differential housing support program must be implemented by considering the area of residence and the actual rent.

Our study is significant in that we conducted an empirical analysis by utilizing data from a certain period after the housing support programs were revised and implemented. However, it is limited in that we could not include various housing support programs in the analysis. In addition, in the case of Korea, there was a limit to the analysis because housing policies such as housing allowance were implemented relatively recently. Therefore, a follow-up study must analyze the effect of not only the public rental housing and housing allowance program but also other housing support programs to demonstrate a general effect

of the housing support programs. Moreover, if follow-up studies perform longitudinal analysis using time series data, they will be able to provide more diverse policy suggestions and implications than those provided in this study.

## 6. Conclusions

The housing problem is a major concern and significantly affects people's lives. Those who cannot afford high housing costs are more likely to be driven out into a poor housing environment, and this negatively impacts housing as well as other areas of their life [3,63]. To solve the housing problem of low-income households, the Korean government has been implementing welfare policies and supporting vulnerable groups by introducing various housing support programs. In recent years, there has been a movement to promote the demand-oriented housing support program due to the limitations of the supply-oriented housing support program. However, few studies have covered this topic. Therefore, this empirical study investigated the effects of the public rental housing program and the housing allowance program on residential satisfaction and the housing cost burden.

The findings revealed that the housing support programs had differential effects on residential satisfaction and housing cost burden based on household attributes. As for public rental housing, the program generally had a positive effect on residential satisfaction. This finding is in line with a previous finding that the supply-oriented housing support program increases residential satisfaction [48,49]. However, the user status of the housing allowance program negatively affected residential satisfaction.

Further, the public rental housing program had a positive effect on reducing the housing cost burden among the whole sample and all household groups. This can be seen as reflecting the reality that tenants ineligible for public rental housing programs are bearing a relatively higher housing cost burden for *jeonsei* or monthly rent than households using the public rental housing. Further, the housing allowance program significantly affected young adult households. Households receiving the housing allowance had a greater housing cost burden than those not receiving the allowance. This means that the amount of housing assistance does not lower the housing cost burden among low-income families to a significant extent. However, these results need not be regarded as a failure of the housing allowance program, but as a basis for revising parts of the program, such as the current eligibility criteria and payment.

**Author Contributions:** Conceptualization, S.K.; methodology S.K.; software, S.K.; validation, S.K.; formal analysis, S.K.; investigation, S.K.; resources, S.K.; data curation, S.K.; writing—original draft preparation, S.K. and J.H.; writing—review and editing, S.K. and J.H.; visualization, S.K. and J.H.; supervision, S.K., J.H. and M.-H.L.; project administration, S.K., J.H. and M.-H.L.; funding acquisition, S.K. All authors have read and agreed to the published version of the manuscript.

**Funding:** This research received no external funding.

**Institutional Review Board Statement:** Not applicable.

**Informed Consent Statement:** Not applicable.

**Data Availability Statement:** The data used in this study come from the Korea Housing Survey. The data can be obtained by visiting the following links: https://mdis.kostat.go.kr (accessed on 9 December 2021).

**Conflicts of Interest:** The authors declare no conflict of interest.

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
