# Peer review of "Effect of Housing Support Programs on Residential Satisfaction and the Housing Cost Burden: Analysis of the Effect of Housing Support Programs in Korea Based on Household Attributes"

_land, doi:10.3390/land11091392_

Round 1
Reviewer 1 Report (Previous Reviewer 1)
The paper is interesting. The authors propose an empirical study on the policy effectiveness of the housing support programs. This study aims to highlight the effects of two housing support programs, namely the public rental housing program and the housing allowance program on residential satisfaction and housing cost burden.
The paper is well-structured, and the authors reach the goal set in this study.
There are still some residual criticisms, I will follow with some suggestions that will improve the quality of this paper.
· As regards the issue of sample size, the authors have partially resolved, as the sample size by type of segment considered is still missing (Whole house-holds, One-person households, Household of a newlywed couple, Households of young adults and Households of the elderly);
· The authors have provided as well as requested a map to locate the case study, from which I deduced that it covers the entire territory of Korea and not the neighbourhoods, please integrate this information in the text of the paper.
· There remains the question of the values of a R2 and Adjust R², perhaps they could be improved by changing the size of the sample. In fact, as for the results highlighted in Table 3, the values for R2 and Adjust R² are acceptable, for those in Table 4 are too low. Perhaps the sample data could be subjected to a preliminary analysis in which to highlight the outliners and converge towards a more stable data structure and a more robust analysis. However, even if it is not possible to make substantial changes it would be correct to give a justification.
Author Response
Please see the attachment.

Reviewer 2 Report (New Reviewer)
General Comments
The study investigates the effects of housing support programmes in Korea, particularly the public rental housing programme and the housing allowance programme, on residential satisfaction and housing cost burden. The paper is well structured and oriented and follows a step by step approach in order to achieve the stated objectives. The research method, consisting of a survey conducted on different types of rental households coupled with the review of policy reports, and the statistical techniques used in the study seem to be appropriate for the theme being investigated. The results of the study are presented and commented upon in an appropriate manner. The “Conclusions” appear to follow the development of the text and some policy implications of the research are drawn.
However, the paper has, in my opinion, some major weaknesses.
1-The research method used in the study needs better elaboration and clarification. In particular, the authors state in Section 3.1 that “...The structuralized survey was carried out from July to December 2020 throughout the country, via face-to-face interviews by trained interviewers….”. Witch survey do the authors refer to? Does it refer to the “Korea Housing Survey? Or does it refer to the survey conducted for this study? In the same line, the phrase “… Rental households were extracted from the data of 51,421 households in the total sample..” needs clarification. What does the total sample refer to? Does it also mean that 51,421 households were contacted for participating in the questionnaire? If not, how many households were contacted for participating in this study?
2- The literature review needs to be expanded. Although the reference works, mostly Korean ones, reviewed for this study are representative of the theme being investigated, some authoritative studies pertaining to housing policies of other countries should be added. I think the international reader would benefit from this approach.
3-The “Discussion” section of the paper also needs better elaboration. I think that the main results of the research should be put in perspective, highlighting the commonalities and differences with prior ones in the literature, both in Korea and international contexts. This aspect also needs to be articulated with the arguments expressed in 2.
Author Response
Please see the attachment.

Reviewer 3 Report (New Reviewer)
Please better describe why you use multiple regression and ordinal logistic regression for your analysis (instead of Evolutionary Polynomial Regression, Geographically weighted regression, etc.).
Author Response
Please see the attachment.

Reviewer 4 Report (New Reviewer)
The paper can make a contribution for the journal because it analyzes the effects of the public rental housing program and the housing allowance program on residential satisfaction and the housing cost burden of policy beneficiaries. Tables and images are properly developed for the paper's contents.
Nevertheless, it still lacks of contents in the theoretical framework and it needs a more international focus for the readership of the journal.
In section "2.1. Representative housing support programs" I suggest to refer to the following works:
- https://www.mdpi.com/2073-445X/11/6/875 because it analyses the qualitative and the quantitative values that influenced the urban form of an urban renewal project and it helps to conceptualize the housing support programs
- On section "2.2 Various factors affecting residential satisfaction and housing cost burden", I suggest to add:
-https://www.sciencedirect.com/science/article/abs/pii/S0264837722000059, because it pinpoints the relevance of housing for the elderly
In section "2.3. The effect of the housing support programs....", add:
- https://www.mdpi.com/2071-1050/14/1/457 because it explains the relation between urban regeneration based on public housing and proximity
- 2020. Alpha City: How London Was Captured by the Super-Rich. London: Verso, as it demonstrates the distortions provoked by capitalist urban regeneration processes.
Based on this, please try to expose better the lessons learned from this paper with a more international view.
Moreover, various parts of the paper are colored in yellow. Please, correct this text.
I require a new version of the paper to check the improvements.
Round 2
Reviewer 1 Report (Previous Reviewer 1)
The authors have responded to all the issues highlighted in the review process. Now the paper can be considered accepted for publication in Land.
Reviewer 4 Report (New Reviewer)
I see the proper advance in the paper. Nevertheless, I also saw that Authors did not take into consideration the relation between urban regeneration based on public housing and proximity (https://www.mdpi.com/2071-1050/14/1/457). I highly suggest to add it briefly. After this integration, the paper can be published.
This manuscript is a resubmission of an earlier submission. The following is a list of the peer review reports and author responses from that submission.
Round 1
Reviewer 1 Report
The paper is interesting. The authors propose an empirical study on the policy effectiveness of the housing support programs. This study aims to highlight the effects of two housing support programs, namely the public rental housing program and the housing allowance program on residential satisfaction and housing cost burden.
The paper is well-structured, and the authors reach the goal set in this study.
Some suggestions can improve the quality of this paper.
It is suggested that
· Select one of the two titles proposed;
· Highlight the paper structure at the end of the Introduction section;
· Improve the description of the area under study;
· Provide a brief introduction to the different types of public rental housing, namely permanent rental housing, national rental housing, long- term Jeonsei housing, and Happy housing, that the authors have recalled at lines 101-103;
· Explain the total sample size and by type of segment considered Whole households, One-person households, Household of a newlywed couple, Households of young adults and Households of the elderly);
· Provide a map of the neighborhoods that have been the subject of analysis;
· Check whether R2 and Adjust R² results could be improved by changing the sample size. In fact, regarding the results highlighted in Table 3, the values for R2 and Adjust R² are acceptable, for those in Table 4, they are too low.
· Integrate in Table 4 the line “Constant”;
· Integrate a Conclusion section;
· Supplement the references.
Reviewer 2 Report
I have a number of issues with this manuscript. I will try and lay them all out in the space below.
1) There are some potentially serious issues with the research design here. Specifying the variables to be used within your model seem flawed. For example, you mention during the introduction that low-income folks are primarily those receiving housing allowance programs. If so, why are you including both variables within your model? I would also think that there is a high degree of correlation between the one-person household and young adult household groups as well. This suggests that there might be some multicollinearity going on. It seems that you employed a strategy of throwing everything but the kitchen sink into the independent variable side of the equation, which is not exactly the best way to form a model.
2) Specifying the variables themselves had some issues as well. For example, what is the way that you defined "housing cost burdened"? It was not altogether clear to me.
3) It was also not 100% clear to me when you were creating variables on your own or taking the variable straight from the housing survey. Because I am not familiar with the survey, it would have been nice to have heard more about that.
4) Isn't it also possible that your research design might have some reverse causation going on here? In other words, isn't it possible that housing cost burden is actually having an effect on the household entering the housing allowance program itself? This is a potentially fatal misjudgment.
5) Similar to #1, it would be best to think about the theoretical correctness of adding variables within your model. For example, why would relations with neighbors and accessibility to parks have an impact on housing cost burden? The theoretical construct behind that association was not adequately explained or buttressed by existing literature.
6) As mentioned before, a little more info about the survey itself would have been helpful here. How many households did you start off with? What were some of the scales that were used? Etc.
7) Was there a reason that you collapsed income into categories rather than using it as a continuous variable? Did the survey itself use categories? If you categorized it yourself, how did you incorporate cutoffs? The same could be said for the housing cost burden variable...how did you create cutoffs? Wass this based off of earlier research?
8) If you collapsed the Likert scales yourself, did you check for reliability of the scale measure? One way to do that is through the Cronbach alpha method.
9) One of your more interesting results was that "the occupancy period negatively affected residential satisfaction" (p. 11). Why do you think that is exactly?
10) I am not entirely sure how all of the subgroups had negative effects of housing allowance on residential satisfaction, and yet the whole sample group effect was positive. How is that possible?
11) On page 15, why would the public transportation access positively affect the housing cost burden?
12) Some of your results were frankly not very rigorous and fairly common sensical. Namely, the following (from pages 15 and 16):
* The housing cost burden was also greater for households whose heads had no other jobs or no jobs at all than those whose heads had office jobs.
* Among the variables of occupancy types, households paying monthly rent were more likely to have a greater housing cost burden than those not paying monthly rent.
* Further, household heads with other jobs or no jobs were more likely to have a greater housing cost burden than household heads with office jobs.
* To sum up, the analysis of housing cost burden in all household groups showed that households living on monthly rent had a greater housing cost burden than those not living on a monthly rent.
These are not exactly riveting findings.
13) Finally, one of your major findings seems problemmatic on its face. On page 19, you state that "households receiving the housing allowance had a greater housing cost burden than those not receiving the allowance. This means that the amount of housing assistance does not lower the housing cost burden among low-income families to a significant extent." It might signal that the progrma itself is a sheer and utter failure. Maybe there is something else going on though...such as people who receive housing allowances take this assistance and stretch themselves out into bigger housing situations than they would have done otherwise. The sheer lack of sense in such a result would give me pause as to stating that the housing allowance really even had such an effect...and would make me radically re-think my modeling.
Reviewer 3 Report
This paper examines the effects of housing support programs, specifically, public rental housing and housing allowance, on residential satisfaction and housing cost burden. It could shed lights on policy making targeting at improving housing welfare of disadvantageous groups. However, the theoretical framework is not well structured, and the research design is flawed. Following are the detailed comments.
1. This study takes residential satisfaction and housing cost burden as the dependent variable respectively and uses the same set of independent variables. It is problematic, as the factors affecting residential satisfaction and cost burden could be different. Besides, the so-called neighborhood attributes are actually satisfaction with the accessibility, which could be taken as measurements (index) for residential satisfaction, rather than the explanatory variables.
2. About the data, it is not clear who are surveyed. The samples are confined to those who are provided either public rental housing or housing allowance? If not, why the proportion of being provided public rental housing (42.8%) and housing allowance (48.7%) so high? And it is not clear whether these two housing support programs are mutually exclusive. Is it possible that one household is provided both public rental housing and housing allowance at the same time?
3. "monthly rent" is misused. First, it is used to refer to housing allowance (page 3), which could be revised into "monthly rent subsidy". Second, it is used to refer housing tenure (renting versus owning) in the model? It seems that the proportion of households paying monthly rent refers to those who are not homeowners? If so, those with "monthly rent"=1 are not eligible for being provided public rental housing, right? In addition, is there any relationship between housing type and "monthly rent"? The four types of housing can be both monthly rented and owned? Introduction on the different housing type is needed.
4. The differentiation of household types is confusing. One-person households, household of newlywed, young adult households, and households of the elderly are not clearly defined. What the criteria of newlywed? Married within certain years? And if a household is composed by only one elderly person, should it be classified into one-person households or households of the elderly?
5. In the manuscript, the authors argue for choosing subjective method to measure housing cost burden. However, it is not convincing. Despite the shortcomings of other methods, the shortcomings of subjective method are also obvious, even worse than the other two. In the data section, the authors state "we selected households with housing cost burdens". Does it mean you eliminated those whose subjective housing cost burden =1 None? Why is that?
6. "We classified those with a household income below 50% of the median income as low-income families". It is inappropriate to define low-income household only within the surveyed samples. It is better to use a standard income classification for reference.
7. The findings are poorly interpreted. "Those who receive the housing allowance are low-income families and are most likely to reside in a relatively poor housing environment. This may be because the subsidy amount that the housing allowance program beneficiaries receive is insufficient for them to move to a better housing environment and improve residential satisfaction." The housing environment has been controlled in the model, so it can not explain the negative effect of housing allowance on residential satisfaction. And this study does not provide evidence on the explanation of the amount of housing allowance.
8. Some content in Table 2 can not be seen.